# Structural changes in noble metal nanoparticles during CO oxidation and their impact on catalyst activity

See Wee Chee [1,2,6], Juan Manuel Arce-Ramos [3], Wenqing Li[3], Alexander Genest[3] & Utkur Mirsaidov [1,2,4,5✉]

The dynamical structure of a catalyst determines the availability of active sites on its surface. However, how nanoparticle (NP) catalysts re-structure under reaction conditions and how these changes associate with catalytic activity remains poorly understood. Using operando transmission electron microscopy, we show that Pd NPs exhibit reversible structural and activity changes during heating and cooling in mixed gas environments containing $O_2$ and CO. Below 400 °C, the NPs form flat low index facets and are inactive towards CO oxidation. Above 400 °C, the NPs become rounder, and conversion of CO to $CO_2$ increases significantly. This behavior reverses when the temperature is later reduced. Pt and Rh NPs under similar conditions do not exhibit such reversible transformations. We propose that adsorbed CO molecules suppress the activity of Pd NPs at lower temperatures by stabilizing low index facets and reducing the number of active sites. This hypothesis is supported by thermodynamic calculations.

[1] Department of Physics, National University of Singapore, Singapore 117551, Singapore. [2] Centre for BioImaging Sciences, Department of Biological Sciences, National University of Singapore, Singapore 117557, Singapore. [3] Institute of High Performance Computing, Agency for Science, Technology and Research, Singapore 138632, Singapore. [4] Centre for Advanced 2D Materials and Graphene Research Centre, National University of Singapore, Singapore 117546, Singapore. [5] Department of Materials Science and Engineering, National University of Singapore, Singapore 117575, Singapore. [6]Present address: Department of Interface Science, Fritz Haber Institute of the Max Planck Society, 14195 Berlin, Germany. ✉email: mirsaidov@nus.edu.sg

To improve the performance of nanoscale catalysts from a fundamental level, we first need to understand the characteristics of their surfaces because catalytic reactions mainly take place on the surface. In this case, the adsorption and oxidation of carbon monoxide (CO) molecules on metal surfaces are important probe reactions for studying surface processes in heterogeneous catalysis[1]. The oxidation of CO to carbon dioxide ($CO_2$) is also an essential part of various industrial processes because CO is a common by-product of incomplete combustion and a toxic gas. In particular, noble metal nanoparticles (NPs), especially those made from Pd and Pt, are ubiquitous catalysts for accelerating this reaction[2]. However, even for these metals, there are unresolved questions about the active state of their surfaces during catalytic conversion. For example, it is known that the surfaces of noble metal crystals change in oxygen ($O_2$)-rich environments, but the effect of these changes on catalytic activity is still being debated[2,3]. To answer these questions, we have to study these materials under reaction conditions because the dynamical working structures may be different from those characterized under vacuum.

The need to clarify these structure–property relationships in catalyst materials has spurred the adaption of several surface characterization techniques[4–7], including infrared (IR) absorption spectroscopy[8], synchrotron X-ray-based methods[9], and X-ray photoelectron spectroscopy[10,11], to enable measurements during reactions and at realistic gas pressures. However, these approaches typically look at bulk single-crystal surfaces, and so, it is not straightforward to extrapolate their results towards the actual behavior of NP catalysts. NPs are known to exhibit structures different from their bulk counterparts[12,13]. Even when supported NPs are used, the measurements are limited to the ensemble behavior of these nanoscale catalysts[14,15], and it is not always clear whether chemical or morphological effects are responsible for the observed changes in spectroscopic signatures. Hence, it is critical that we clarify the detailed morphology of individual NP catalysts under these same reactive environments in order to correlate a catalyst's structure with its activity.

Transmission electron microscopy (TEM) has always been a powerful technique for studying NP catalysts because of its high spatial resolution and complementary spectroscopic capabilities[4,16,17]. While the samples are typically characterized under vacuum in the TEM column, similar advances in instrumentation have enabled in situ studies at realistic reaction conditions[18,19]. TEM holders with microfabricated nanoreactors[20–22] now allow us to capture the morphology of NPs during catalytic reactions at atmospheric pressure and elevated temperatures. So far, this approach has been used to follow the oscillatory behavior in Pt NPs during CO conversion[23] and study the effect of CO adsorption on Pt NPs[24]. Furthermore, the integration of mass spectrometry and microcalorimetry allows us to track the reaction kinetics with these in situ setups[23].

Here, we use TEM to follow the morphological changes in noble metal NP catalysts, namely Pd, Pt, and Rh, under mixed gas environments containing $O_2$ and CO as a function of temperature, while tracking their catalytic conversion with inline mass spectrometry. Pd exhibits the most unexpected structural transformations. First, we observe that Pd NPs restructure and form extended low index facets when CO is added into the $O_2$-rich gas stream at 200 °C. In this form, the Pd NPs are not active toward CO oxidation up to ~400 °C. Second, above 400 °C, the Pd NPs become rounded. Gas composition measurements indicate that this transformation is accompanied by the initiation of the CO oxidation reaction. Third, the low index facets reform when the temperature is lowered back below 400 °C, and the NPs become inactive again. Hence, our observations indicate that an inactive-to-active transition occurs concurrently with the destabilization of

the low index facet-dominated structure. Pt and Rh NPs under the same conditions do not exhibit such reversible structural transformations. Using density functional (DFT) calculations and Wulff constructions, we show that when in contact with a binary $CO/O_2$ mixture, the number of under-coordinated edge sites on a Pd NP abruptly increases at temperatures above 300 °C, whereas the number of edge sites on a Pt NP remains relatively constant with temperature. These results provide direct evidence of how the morphology change due to CO adsorption can suppress the number of active sites on a Pd NP's surface at lower temperatures. The reversibility of these surface facet transformations in Pd NPs with temperature also implies that the active morphology of such catalysts cannot be observed outside of reaction conditions.

## Results

**Re-structuring of Pd NPs at 760 Torr of 9% CO, 18% $O_2$, 73% He**. Figure 1a shows a schematic of the experimental setup where metal NPs are encapsulated within a microfabricated gas cell with a thin film heater. Using this approach, we obtained image sequences of a NP catalyst's structure under 760 Torr pressure of reactant gases and at elevated temperatures (Fig. 1b). Concurrently, the CO, $O_2$, and $CO_2$ partial pressures in the outlet gas line were monitored with an inline mass spectrometer (Fig. 1c). It is known that the energetic electrons used in TEM can introduce artifacts to the observed phenomena in in situ experiments, and it has been reported that an electron flux of below $200e^-/(Å^2 s)$ is required to avoid these effects for gas-phase studies[23,25]. To minimize any effects the electron illumination may have on the reaction, we chose to image the NPs at only ~$100e^-(Å^2 s)$. However, imposing such a stringent criterion in these gas-phase experiments also results in noisy images that have degraded resolution and contrast due to poor signal-to-noise ratios[26]. To mitigate the signal-to-noise ratio issue, the image sequences were captured using the electron counting mode of a direct electron detection camera, an imaging mode used in cryo-electron microscopy of biological specimens[27], and the imaging parameters were optimized to maintain lattice resolution.

Initially, the NPs were heated to 200 °C in an environment containing 80% He and 20% $O_2$. The 1st image in Fig. 1b shows that the NP was relatively round under these conditions. Next, CO gas was added into the gas stream (2nd image in Fig. 1b), bringing the overall gas composition to 73% He, 18% $O_2$, and 9% CO. The ratio of partial pressures $\left(\frac{P_{CO}}{P_{O_2}}\right)$ is equal to 0.5. The NP restructured when CO was introduced. Figure 2a is a magnified image of the same NP at this stage. The NP surface consists of mostly flat low index facets as indicated by the dashed lines in the TEM image, and these facets further terminated in sharp corners. The fast Fourier transform (FFT) from the NP image indicates that the lattice fringes can be indexed to the {111}, {200}, and {220} planes of metallic Pd. A section of the vicinal facet on the lower right of the NP is enlarged in the bottom inset of Fig. 2a where it shows the formation of sawtooth steps that are made up of smaller {111} facets. The NP maintained this faceted morphology when the temperature was increased to 300 °C (Fig. 1b). Mass spectrometry measurements indicate that there was no significant conversion of CO to $CO_2$ at temperatures between 200 and 300 °C. This behavior is consistent with CO-poisoning of the catalyst surfaces where the strong binding of CO molecules to the surfaces displaces $O_2$ molecules and prevent the oxidation reaction from occurring[2].

The next three images in Fig. 1b show how the surface structure of this NP changed as we further varied the temperature from 300 to 600 °C and back (see Supplementary Fig. 2 for the corresponding temperature profile). The NPs became active after the temperature was raised to 500 °C, and we

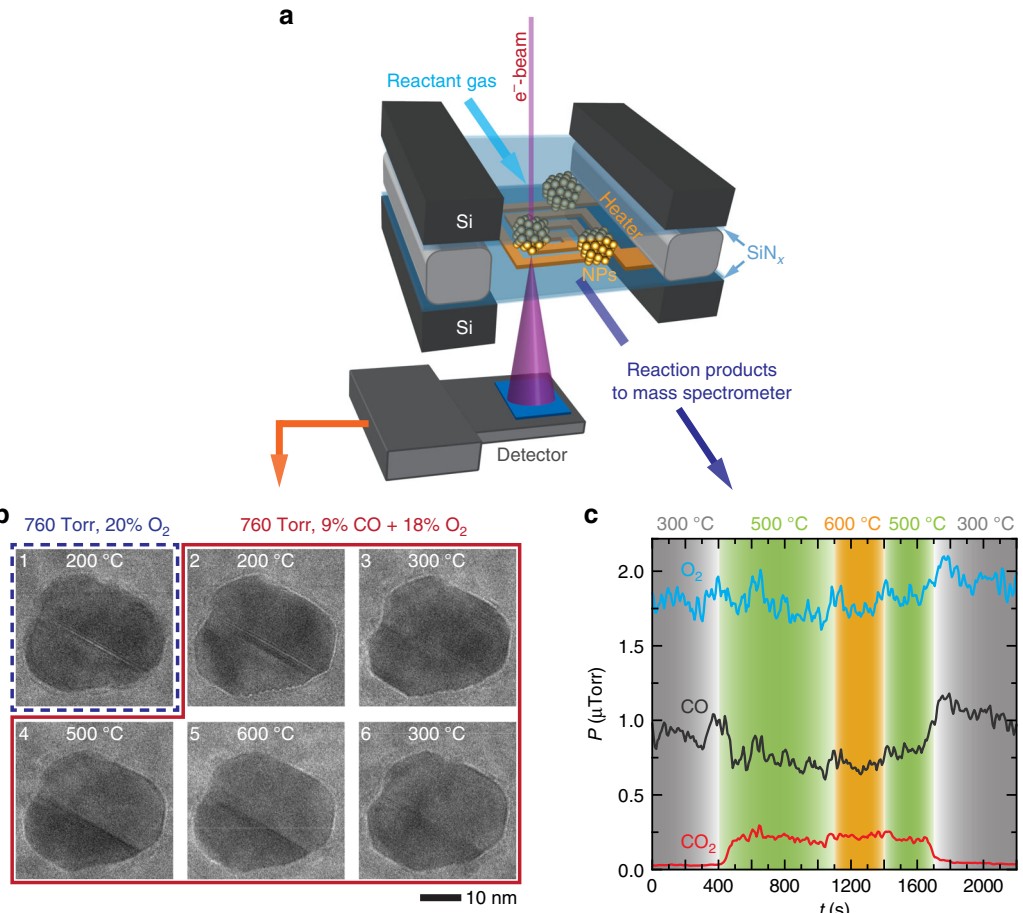

**Fig. 1 Operando transmission electron microscopy (TEM) of Pd nanocatalysts during CO oxidation. a** Schematic of the experimental setup where metal nanoparticles (NPs) are synthesized within a microfabrication gas cell with an integrated thin-film heater. The reactant gases after passing over the catalysts are analyzed using an inline mass spectrometer. For enlarged images of panels 2, 5, and 6, see Supplementary Fig. 1a. **b** Sequence of TEM images of a Pd NP under different gas environments and temperatures. **c** Changes in gas composition (carrier gas is He) obtained from the inline mass spectrometer during temperature changes between 300 and 600 °C. These results correspond to panels 3–6 in the red box shown in **b**.

saw a significant increase in $CO_2$ production (Fig. 1c). More importantly, the change in activity occurred concurrently with a change in a NP shape, where the flat facets and sharp corners became more rounded. The sawtooth steps, previously seen at the vicinal facet (Fig. 2a), also disappeared. A higher magnification image of the NP at 500 °C is shown in Fig. 2b. Figure 2c shows an image sequence extracted from the movie that was recorded during the temperature ramp from 300 to 500 °C at a rate of 2 °C/s (Supplementary Movie 1). The measured temperature and heater power during this ramp are plotted in Fig. 2d. At around 450 °C, there was a spike in the measured temperature, and the temperature control software reduced the heater power to maintain the ramp rate of 2 °C/s. The oxidation of CO to $CO_2$ is an exothermic reaction, and so, the heat released by the reaction raises the temperature, which further catalyzes the oxidation reaction. This behavior is commonly known as ignition[28]. The observed spike in temperature (Fig. 2d) matched well with the time where the NP changed its shape in the movie (Fig. 2c). Therefore, our observations suggest that the shape change correlated with the ignition of the CO oxidation reaction. The NP morphology and the CO conversion rate did not appear to change with a further increase to 600 °C. Panel 6 in Fig. 1b shows that lowering the temperature back to 300 °C led to a reforming of flat low index facets in the NP, indicating that the surface transformation was reversible (Supplementary Movie 2).

Together with the restructuring, we can see a commensurate drop in CO conversion (Fig. 1b), which means that the NPs became inactive again. For clarity, enlarged images of the NP at the key temperatures of 200, 500, and 300 °C are shown in Supplementary Fig. 1a. Images of additional Pd NPs acquired in different experiments but under the same reaction conditions are shown as Supplementary Figs. 3 and 4 to illustrate that this behavior is general.

More significantly, our observations show that the structural transformation occurred concurrently across all the NPs in the field of view. Lower magnification movies capturing such behavior in multiple NPs are provided as Supplementary Movies 3 and 4 for temperature ramps from 300 to 500 °C and from 500 to 300 °C, respectively. Clearly, the observed stabilization and destabilization of low index facets are reproduced across different Pd NPs. These movies, which are challenging to acquire due to thermal drift during temperature changes, also provide unambiguous evidence that the restructuring occurs together with the ignition peak seen in the microcalorimetry measurements (such as that seen in Fig. 2d). To further demonstrate that the observed facet stabilization indeed is due to CO adsorption, we include in Supplementary Fig. 5, images from a control experiment that compared the same NP in a $N_2$ environment and in a $CO + N_2$ environment at 200 and 600 °C. The experiment indicates that the low index facets can be dynamically stabilized in the presence of

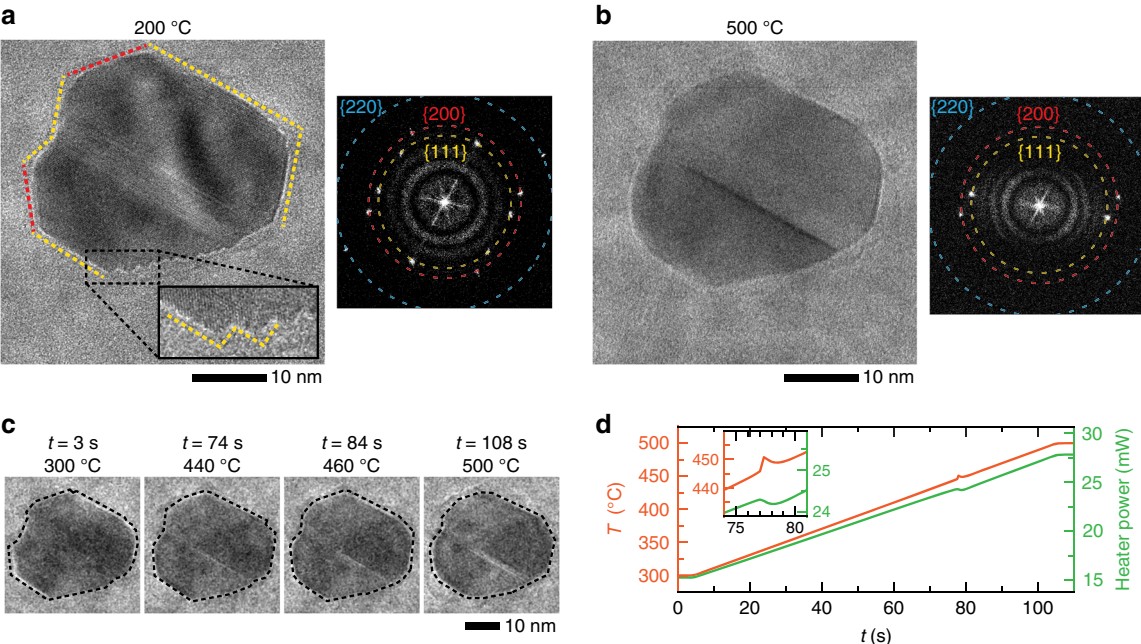

**Fig. 2 Detailed results for the experiment described in Fig. 1. a** Higher magnification images of the NP in Fig. 1b at 200 °C and under 760 Torr of 9% CO, 18% O$_2$, and 73% He $\left(\frac{P_{CO}}{P_{O_2}} = 0.5\right)$, and its corresponding fast Fourier transform. The low index facets are highlighted with dashed lines of different colors that correspond to reflections in the Fourier transform. Inset further magnifies the step structure at the vicinal facet on the lower right of the NP. **b** Higher magnification image of the NP after heating to 500 °C under 760 Torr of 9% CO, 18% O$_2$, and 73% He gas. Notice the absence of sharp corners in the NP and the steps on the vicinal facet. **c** Image sequence extracted from a movie recorded during the temperature ramp from 300 to 500 °C (Supplementary Movie 1). The sequence shows the NP changing from a faceted morphology to a rounded one as the temperature increases. **d** Plots of the measured temperature (orange curve) and heater power (green curve) during the temperature ramp of 2 °C/s. The inset highlights a spike in the measured temperature and the corresponding reduction in heater power needed to compensate for the exothermic release of heat from the NPs (y-axes have been re-plotted for the inset to accommodate both curves). For enlarged images of **a** and **b**, see Supplementary Fig. 1a.

CO up to 600 °C. In contrast, heating the NPs only in N$_2$ results in the desorption of the adsorbed CO molecules, and the NP became round.

**The restructuring of Pd NPs at higher CO to O$_2$ ratio.** The next question that needs to be addressed is if the changes in CO$_2$ production were simply a temperature effect or if there was an additional contribution from the structural transformations observed in the Pd NPs. The inactivity of catalysts at low temperatures is usually associated with surface poisoning by adsorbed CO molecules displacing O$_2$ molecules[2]. Previous studies with bulk Pd single crystals had shown that increasing the partial pressure of CO increases the ignition temperature[28,29]. This shift in the ignition temperature can be explained by an increase in temperature needed to desorb CO molecules from the metal surface due to higher CO partial pressure. Hence, if Pd NPs also restructure at a higher temperature with an increased CO partial pressure, it will support a hypothesis that the changes in CO$_2$ production are not solely a temperature effect, but are also associated with the structural transformations caused by CO adsorption/desorption.

To see if we can further stabilize the low index facets in Pd NPs by increasing CO partial pressure, we performed operando TEM experiments at a CO to O$_2$ ratio of $\frac{P_{CO}}{P_{O_2}} = 1.6$ (64% He, 16% O$_2$, 20% CO). The results are summarized in Fig. 3, where the measured temperature and heater power profiles are shown in Fig. 3a, mass spectrometry data in Fig. 3b, and the image sequence of a Pd NP at three temperatures in Fig. 3c. The first image in Fig. 3c shows that at 400 °C, the NP still has flat low index facets and sharp corners. In this case, we had to raise the upper limit of

the temperature ramp to 600 °C to see the corners of the NP become rounded (Supplementary Movie 5), and again there was a corresponding increase in CO$_2$ production (Fig. 3b). Lowering the temperature back to 400 °C (Supplementary Movie 6) caused a drop in CO conversion and the low index facets to re-form. Additional sets of images comparing the same NPs at the two CO to O$_2$ ratios of 0.5 and 1.6 (Supplementary Fig. 6) indicate the same trend in structural evolution. The microcalorimetry measurements during these experiments indicate that the ignition temperature had indeed shifted upward from ~420 °C at $\frac{P_{CO}}{P_{O_2}} = 0.5$ (Supplementary Fig. 7) to ~500 °C at $\frac{P_{CO}}{P_{O_2}} = 1.6$ (Fig. 3a). In another experiment at $\frac{P_{CO}}{P_{O_2}} = 1.6$, we also searched for Pd NPs on the vertical edge of the SiN$_x$ window, which allow us to visualize the re-faceting from a different perspective. Supplementary Fig. 8 shows images of one such Pd NP where we see its side and top facets. Consistent with the results described above, we observed the development of low index facets on the top of the NPs at lower temperatures. From these experiments, we conclude that the structural transformation induced by the adsorption of CO molecules also renders the Pd NPs inactive at temperatures below their ignition temperature.

**Comparing noble metal NPs under similar reaction conditions.** We highlight here that the behavior we observed in Pd NPs is, in fact, opposite to that reported for Pt NPs[23,24]. Earlier studies show that the adsorption of CO at lower temperatures leads to a surface reconstruction of the Pt {100} facets into stepped higher Miller index facets[24] and Pt NPs were reported to be more faceted when active[23]. To confirm the difference in behavior, we

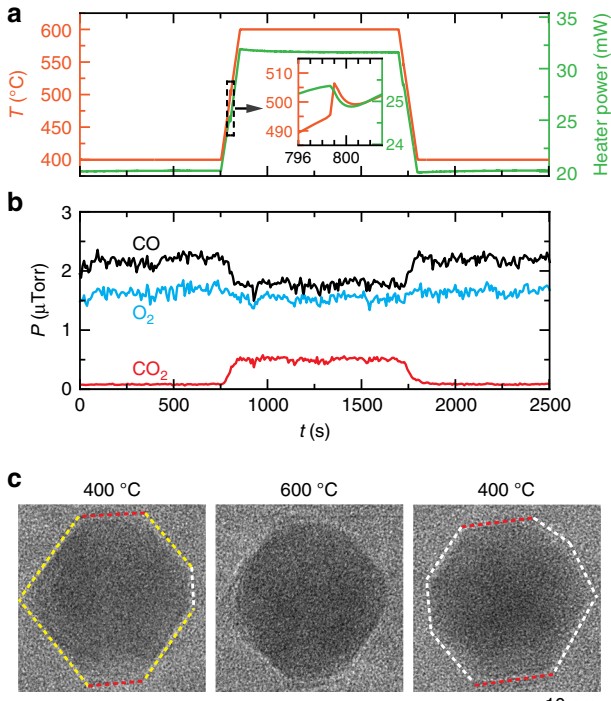

**Fig. 3 Pd NPs imaged in a gas environment with CO to $O_2$ ratio of 1.6.**
**a** Measured temperature and heater power profile from the experiment. The inset highlights the temperature spike around the ignition point and the corresponding reduction in heater power expanded from the region demarcated by the dashed box (y-axes have been re-plotted for the inset to accommodate both curves). **b** Corresponding measurements of the CO, $O_2$, and $CO_2$ content in the outlet gas stream during the experiment. **c** Image sequence describing the morphological changes in a Pd NP at different temperatures in the experiment. Dashed lines highlight the outline of the NP where yellow lines correspond to {111} facets, red lines to {100} facets, and white lines to facets that cannot be indexed from the individual images' fast Fourier transform.

performed a comparative study of Pt and Rh NPs under similar conditions. The two metals exhibited behavior different from Pd and from each other. Pt NPs did not show obvious structural changes with increasing temperature, whereas Rh NPs reduced from an initial oxide state to metal with the introduction of CO into the gas mixture. Figure 4 summarizes these results with three image sequences describing the morphological changes in a ~10-nm NP of each type in an $O_2$-rich environment with $\left(\frac{P_{CO}}{P_{O_2}}\right) = 0.5$ at different temperatures.

Figure 4a shows a Pd NP with almost atomically flat {100} and {111} facets, which terminate in sharp corners, at 200 °C (inactive). The enlarged image of this NP can be found in Supplementary Fig. 1b. Consistent with the results presented earlier, the NP became more rounded at 500 °C and above (active), and the facets re-formed when we decreased the temperature back to 300 °C (inactive). The mass spectrometry data exhibited trends similar to that shown in Fig. 1c, which again indicate that the faceted NPs were inactive toward CO oxidation.

Pt NPs re-structured more subtly as represented by Fig. 4b. Supplementary Fig. 9 shows additional Pt NPs to illustrate that the small changes in the structure were general across different NPs. When CO was introduced to the $O_2$-rich environment at 200 °C, the NPs seemed to slightly change their shape, but maintained a rounded appearance. This behavior is consistent with that reported by Avanesian et al.[24] where CO introduced to Pt NPs in a $N_2$

environment at 250 °C only led to the formation of stepped higher index facets on the {100} facets. The authors proposed that the restructuring created more under-coordinated sites on the Pt NP surface, which may account for a small increase in the $CO_2$ partial pressure seen in the mass spectrometry data at 300 °C (Supplementary Fig. 10). The conversion of $CO_2$ continued to increase as we further raised the temperature to 500 and 600 °C (Supplementary Fig. 10). At the higher temperatures, the Pt NPs appeared to be somewhat faceted (Fig. 4b and Supplementary Fig. 9), but these changes are not as obvious as that seen in the Pd NPs (Fig. 4a). These results suggest that the increase in CO conversion in Pt NPs was an effect of increased kinetics at higher temperatures.

For Rh, lattice fringes in the NPs indicated that they could be found in both oxide and metal forms at 200 °C. The lattice fringes in the first image shown in Fig. 4c have a *d*-spacing of ~0.262 nm and is a possible match for $Rh_2O_3$[30] (Rh (111) has *d*-spacing = 0.222 nm[30]). The introduction of CO led to the gradual reduction of the oxide towards metallic Rh. A slow increase in $CO_2$ production was also seen when the temperature was raised to 300 °C (Supplementary Fig. 11). An ignition peak was observed when the temperature was raised just above 300 °C (Supplementary Fig. 11). There was no significant change in the NP structure and $CO_2$ production between 500 and 600 °C. Lowering the temperature from 500 °C caused a drop in $CO_2$ production, but no oxidation of the NP was apparent from the images. This behavior is consistent with a de-stabilization of the oxide phase in the presence of CO, as seen at the start of the experiment.

## Discussion

These operando TEM observations allow us to associate the structure of NPs with their catalytic activity and understand how the changes in NP structure can influence the availability of active sites on a NP's surface. Clearly, Pd NPs showed the most interesting behavior with a sharp structural transformation that was accompanied by an inactive to active transition. Another striking feature in these observations is the relatively high temperature required for CO oxidation to take place in Pd. The ignition temperature we found for Pd NPs was higher than that reported for bulk Pd crystals at similar CO to $O_2$ ratios by more than 100 °C[29]. In comparison, Rh and Pt NPs started to convert $CO_2$ at 300 °C, which is closer to that of their bulk counterparts[29]. We emphasize here that the $SiN_x$ membrane support does not contain significant oxygen and should not exhibit the catalyst–support interactions that are common to oxide supports[31,32]. Therefore, we expect our results to be representative of the intrinsic behavior of these metal NPs.

There are two possible explanations for the inactive to active transition observed for Pd NPs. First, the stabilization of the low index facets in the NPs results in a decrease in the number of under-coordinated sites on the NP surface. It is well-accepted that these under-coordinated sites are critical for catalytic activity[31]. The relationship between these sites and reactivity has also been verified with theory[33,34]. This explanation is further supported by the loss of the stabilized morphological features after the ignition of the CO oxidation reaction. The rounding of the NPs can be explained by the desorption or oxidation of CO molecules, which then exposes more under-coordinated active sites for the oxidation reaction to proceed on. Second, the behavior is due to oxidation or reduction of the metallic surface. It is known from previous work that two forms of oxides, a thin epitaxial surface oxide and a bulk oxide, can form on the Pd crystal surface under reaction conditions[2]. It was reported that the surface oxide can have at least as much activity as the metallic surface, but the thicker oxide is less active[3]. In the larger Pd NPs, we can see what appears to be a bulk oxide (Supplementary Fig. 12) based on their measured lattice fringes, which were ~0.204 nm. This spacing is not present in metallic Pd[30] and is

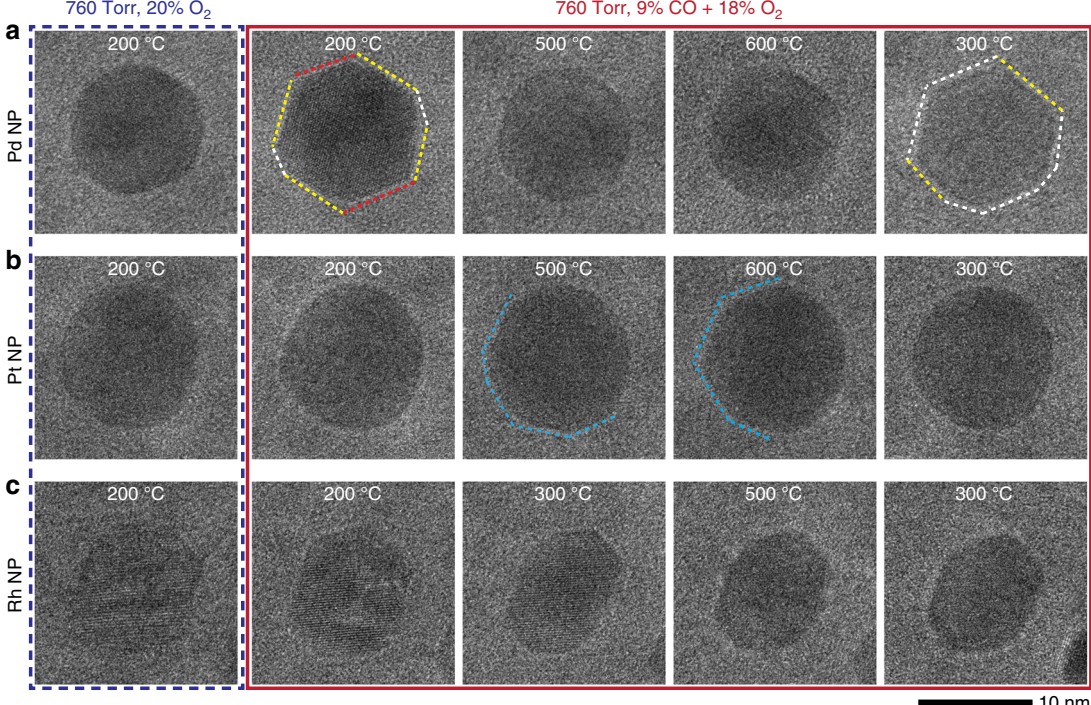

**Fig. 4 A comparison of Pd, Pt, and Rh NPs during CO oxidation.** TEM image sequences of ~10-nm **a** Pd, **b** Pt, and **c** Rh NPs under similar reactions conditions $\left(\frac{P_{CO}}{P_{O_2}} = 0.5\right)$. The Pd NP exhibits the same behavior as other NPs described so far in the manuscript (enlarged images of the NP at 200, 500, and 300 °C are provided in Supplementary Fig. 1b). Dashed lines highlight the outline of the Pd NP where yellow lines correspond to {111} facets, red lines to {100} facets, and white lines to facets that cannot be indexed from the individual images' fast Fourier transform. The Pt NP only showed subtle changes in its morphology. Light blue dashed lines indicate possible faceting on the Pt NPs at higher temperatures. The Rh NP changes from a faceted oxide NP to a more rounded metallic NP as it reduced oxide to metal over the course of the experiment. The FFTs shown in Supplementary Fig. 11a indicate the presence of both {100} and {111} facets in the reduced state.

close to the *d*-spacing of PdO (102)[30]. However, these oxides did not exhibit significant evolution during temperature changes, and so, we ruled out the possibility that they were responsible for the inactive-active transition. The role of the surface oxide is more challenging to determine, especially with the weak contrast of oxides in these microfabricated reaction cells and under our low electron flux imaging conditions. For the NP shown in Fig. 1, we can see a slight deviation from the lattice spacings of metallic Pd, in one of the three pairs of (111) spots and the (020) spots in the FFT. This deviation translates to a lattice expansion of at most 0.01 nm, which may suggest surface oxidation. However, this deviation was not present anymore at 300 °C, a temperature at which we still see no activity. The lattice fringes at higher temperatures were consistent with metallic Pd. However, it had been challenging to find other NPs where we could track the lattice fringes through different temperatures due to the shift of NP orientation with heating and cooling (described further in Methods). At present, we are unable to make a definitive conclusion about the role of the epitaxial surface oxide based on the current results.

To verify whether the change in activity can be explained by a change in the number of active sites, we modeled Pd and Pt NPs with diameters of ~13 nm using a Wulff construction to predict the shape of the clusters while interacting with a CO and $O_2$ gas mixture. DFT calculations were carried out with the BEEF-vdW functional[35] to model the adsorption at the low index facets of Pd and Pt, i.e., {111}, {100}, and {110}. The Fowler−Guggenheim (F−G) isotherm was used to model the adsorption of the binary mixture in the spirit of previous studies[36–38]. Following this scheme, we predicted the required interface tension energy at the partial pressures of 68 Torr for CO and 136 Torr for $O_2$ and temperatures from 200

to 600 °C. Here, we briefly summarize our results, which are discussed in detail in the Supplementary Note. The shape of Pd NPs changes drastically from a {110} faceted to a multi-faceted particle in a temperature interval smaller than 50 °C, starting at 275 °C. This change in shape and facet exposure for Pd leads to an increase in the fraction of under-coordinated edge sites from 5% of its total surface sites at 275 °C to 11% at 300 °C, and 12% at 325 °C. In contrast, a Pt NP modeled at the same partial pressures of CO and $O_2$ exhibits only a gradual change in shape over the whole interval of temperatures, 200–600 °C. The fraction of edge sites increases by only 2%, starting from 8% at 200 °C to 10% at 600 °C. Given the typical accuracy of DFT calculations, one cannot expect a perfect match between the predicted and measured temperature at which Pd NPs start to change shape. However, the qualitative trend in modeling matches in a quite impressive fashion with the experiments. Hence, these thermodynamics calculations rationalize the proposed mechanism for Pd NPs, where a sharp increase in the number of under-coordinated sites that occurs concurrently with CO desorption is responsible for the abrupt rise in CO conversion at elevated temperatures. Whereas for Pt NPs, the incremental increase in conversion rate with temperature is largely the result of increased kinetics at higher temperatures.

The greater reactivity of the NP corners and edges relative to the flat facets is generally understood in terms of the stronger binding of CO molecules to the under-coordinated sites at those locations[34]. It is known from earlier IR spectroscopy studies that CO adsorbs more strongly to the edge and kink sites compared to the Pd surfaces at high CO coverages[39–41]. Previous DFT calculations also confirmed that the adsorption of CO molecules is strongest at these edge features, which is stronger than that on

regular {100} and {111} facets[40]. Similar calculations using our current models also confirm this stronger binding of both CO and O. For CO adsorption to a hollow site near the edge to the (100) surface, we obtain a binding energy of 1.79 eV compared with 1.55 eV for adsorption to a central hollow site in an extended Pd (111) surface. The binding energy of O is 1.17 eV near the edge compared with 1.09 eV on the extended surface. These values are consistent with the trends reported in earlier calculations by Falsig et al.[34].

We also highlight that our observed NP geometry, where Pd NPs expose predominantly {100} and {111} facets, is, in fact, commonly reported in previous vibrational spectroscopy studies. In general, the spectroscopic signatures of CO molecules adsorbed on oxide-supported Pd NPs obtained from IR spectroscopy[39–43] and vibrational sum frequency spectroscopy[40,44,45] had been found to reproduce those obtained from (100) and (111) single crystal surfaces. This phenomenon is usually interpreted as the NPs adopting a thermodynamically favorable structure during heat treatment, but our results now reveal that it was the adsorption of CO that had instead altered the surface structure of these Pd NPs. To the best of our knowledge, only Hicks et al.[42] had proposed based on data from temperature-programmed desorption and IR spectroscopy measurements that the atoms in Pd NPs can re-arrange during CO adsorption, and this behavior has not been verified until now. Moreover, our results clearly show that the rearrangement of surface atoms extends to the adsorption of CO molecules under $O_2$-rich conditions.

The observed restructuring of Pd NPs can be explained by the surface diffusion of CO molecules toward the edges bringing about the motion of the surface atoms on a Pd NP and adding them to the corners and edges. Recent pump-probe measurements indicated that CO molecules indeed diffuse towards NP edges[46]. Furthermore, it has been shown that the adsorption of CO molecules at room temperature can lead to the motion of Pd adatoms on a $Fe_3O_4$ surface via the formation of Pd carbonyls[47]. The contrasting morphological evolution between Pd and Pt NPs can also be explained by how CO molecules bind differently to the respective metal surfaces. IR spectroscopy measurements had indicated that CO molecules bind predominantly to the terrace and step sites of Pt[48,49], and not the edges sites. We can further contrast Pd and Pt NPs using a recent paper that show detailed operando TEM observations of how Pt NPs evolve under thermally cycled CO oxidation reaction conditions by Plodinec et al.[50] Similar to the results presented here, their results indicate that the conversion of CO start below the ignition temperature of Pt NPs. They further describe a largely irreversible transformation of Pt into a thermo-dynamically stable structure with low index facets over time, which gradually de-activates the NPs. On the other hand, we report reversible facet transformations for Pd NPs during thermal cycling (Supplementary Fig. 6) that indicates a dynamical phenomenon that is dependent on the gas environment and temperature. In short, our results suggest a de-activation mechanism where the adsorption of CO molecules on a Pd NP at relatively low temperatures not only poisons the surface, but also drives a change in the NP morphology that further suppresses the population of active sites.

Although previous in situ TEM studies have indicated that Pd NPs can change their shape in single-gas environments at atmospheric pressures, namely $N_2$[51], $O_2$[52], and $H_2$[52], the dynamical restructuring of Pd NPs under reactive gas conditions has not been reported yet. More importantly, our observations of reversible Pd NP facet transformations at different temperatures has significant implications for other investigations aiming to elucidate the active structures of catalysts. Since the Pd NPs revert into an inactive structure with decreasing temperature, it means that we cannot extrapolate the active NP structure from Pd catalysts that are studied outside of reaction conditions, at least for

the CO oxidation reaction. Hence, our results also provide mechanistic insight for interpreting existing catalytic studies of NPs because these structural dynamics can complement earlier work that only have access to the ensemble properties of the NPs[39–43,53]. For example, it may shed light on contradicting results involving the adsorption of CO on small Pd NPs where some studies reported enhanced binding, whereas others did not[32]. This discrepancy had been attributed to the different oxide supports used[32]. We speculate that the key to understanding this phenomenon lies instead in clarifying the effect(s) of process gas conditions or in looking at how the supports disrupt the facet stabilization caused by CO molecules.

In conclusion, using operando TEM, we have unambiguously established that Pd NPs exhibit an unexpected reversible structural transformation in oxygen-rich gas environments that contain CO during heating and cooling. Below 400 °C, the adsorption of CO molecules causes the growth of low index facets. This CO-stabilized structure contributes to a reduction in the activity of Pd NPs below their ignition temperature, which is in addition to CO poisoning. The NPs only become active for CO oxidation when this structure is disrupted by heating to higher temperatures, and the NP surfaces become rounded. This restructuring occurs concurrently with the ignition of the oxidation reaction. Furthermore, this structural transformation reverses when the temperature is lowered, resulting again in inactive NPs. Similar behavior is not seen in Pt or Rh NPs. Our modeling results indicate that the reduced activity of Pd at lower temperatures can be explained by the stabilization of low index facets causing a reduction in the number of under-coordinated sites on the NP, sites that are responsible for the catalytic activity. These results describing the changes in NP morphology under different experimental conditions show how gas environments can alter the structure of NPs, even at relatively low temperatures and have important implications for our under-standing of how CO adsorption can affect NP surfaces. Further work involving operando TEM observations of oxide-supported NPs, for example, should allow us to unravel the mechanisms behind the surface reactivity changes of NPs in more complex systems. In fact, we believe that the path to developing Pd-based NP catalysts with high activity at low temperatures may lie in understanding how supports or alloying interfere with the facet stabilization effect of CO molecules that we have described. Lastly, this work highlights the importance of investigating catalyst structures under reaction conditions.

## Methods

**Synthesis of Pd, Pt, and Rh nanoparticles**. We modified the procedure described by Vendelbo et al.[23] to synthesize the NPs in our experiments. For the Pd NPs, a 1 µL droplet of 10 mM sodium tetrachloropalladate (II) ($Na_2PdCl_4$, Cat. No. 205818-1G, Sigma-Aldrich Co., St Louis, MO, USA) aqueous solution was pipetted onto the microfabricated chip with the heating element (DENSsolutions, Delft, Nether-lands). This solution droplet was allowed to dry on the chip. Then, the chip was loaded into a DENSsolutions Climate holder (DENSsolutions, Delft, Netherlands) and heated to 500 °C under vacuum. Next, the chip was removed from the holder and rinsed in deionized water to remove the residual salt that was left on the chip. This procedure is crucial for ensuring that new NPs will not form in the subsequent experiments and the as-formed NPs are largely limited to the heating element. Using the same approach, the Pt NPs were synthesized with an aqueous solution of 10 mM hydrogen hexachloroplatinate (IV) hexahydrate ($H_2PtCl_6·6H_2O$) (Cat. No. 206083, Sigma-Aldrich Co., St Louis, MO, USA), whereas the Rh NPs were syn-thesized with a 1 mM aqueous solution of the hydrated salt, $RhCl_3·xH_2O$ (Cat. No. 206261, Sigma-Aldrich Co., St Louis, MO, USA). Then, the entire gas cell was reassembled, and leak checked before loading into the TEM. The NPs synthesized in this manner had a broad distribution of sizes and shapes. To ensure that the NPs achieved a stable morphology prior to experiments, the NPs were activated by an initial flow of 9% CO, 18% $O_2$, and 73% He, and heating to 600 °C for at least 30 min. This second treatment resulted in NPs of different sizes that were well-separated from each other (Supplementary Fig. 13). Low magnification images collected periodically during the subsequent experiments confirmed that further coalescence of the NPs did not occur (Supplementary Fig. 13).

**Operando TEM**. The TEM studies were performed in a Thermo Fisher 300 kV Titan TEM. Images were collected with a Gatan K2-IS direct electron detection TEM camera (Gatan Inc., Pleasanton, CA, USA). In these experiments, we optimized the imaging conditions such that the electron fluxes were kept to ~100 $e^-$/(Å$^2$ s) at all times to avoid electron beam-induced artefacts. These fluxes are lower than what's commonly used in similar high resolution in situ TEM studies[22,23,54]. To acquire the movies, a screen capture program within the Gatan Microscopy Suite (GMS) software was used to extract image sequences at 5 frames/s.

For operando TEM experiments, the NPs were first heated to 200 °C under a flow of 20% O$_2$ and 80% He for 30 min before the introduction of CO. The gas composition was adjusted by changing the gas flow in individual mass flow controllers installed within the gas delivery system (DENSsolutions, Delft, Netherlands). Here, two gas cylinders were used, one with pure CO and one with a pre-mixed gas composition of 20% O$_2$ and 80% He. To increase the residence time of gas molecules on the NPs, a flow rate of 0.07-0.09 mL/min in the gas cell was used. The inline gas analyzer (DENSsolutions, Delft, Netherlands) was connected to the holder outlet line, and the gas compositions were measured with a quadrupole mass spectrometer (Stanford Research Systems, Sunnyvale, CA, USA). A leak valve controlled the amount of gas going into the analyzer chamber, which was adjusted to allow for a chamber pressure in the range of $10^{-5}$ Torr. We mention here that due to the relatively long duration of our experiments, there was a build-up of He in the analyzer chamber over time. We suspect that this build-up was the cause behind the drift seen in our spectrometry data.

Our preliminary experiments indicate that the most important temperature range for observing dynamical changes in NP morphology during the CO oxidation reaction is between 300 and 500 °C. To capture the transformations within this interval, we programmed the heater to ramp the temperature at a rate of 2 °C/s, and movies were acquired during the heating program. The increase (decrease) in temperature resulted in a buckling of the thin SiN$_x$ membranes as they expand (contract), which caused thermal drift in the lateral and vertical position and orientation of the NPs during image recording. The drift in position and change in height were manually corrected on-the-fly to keep the NPs within the field of view and roughly in focus. The images were further drift-corrected during the post-processing of the movies. The tilt of the NPs, however, cannot be corrected easily, and so, the lattice fringes that could be seen in the same NP changed at different temperatures. Therefore, the FFT patterns also changed correspondingly. We further mention here the images presented in the manuscript had been extracted from a pool of several experiments, each using a new nanoreactor. In all cases, NPs of the same material behaved consistently across the separate experiments.

**Image and data processing**. The static images presented in the figures were extracted from image stacks that contained 80 electron-counted images acquired at an exposure of 0.025 s per image. First, the individual images in the stack were drift-corrected and summed to a cumulative exposure of 1 s (40 frames) using the GMS software. Then, the summed images were processed using a median filter with a structuring element of 3 pixels × 3 pixels in size to reduce their salt-and-pepper noise.

The drift-corrected movies were averaged by five and compressed using a XVID codec to reduce them to a size suitable for uploading.

The profiles in mass spectrometry results shown in the figures and supplementary figures had been processed using a 5-point average.

**Calculations of equilibrium shapes for NPs**. The models of metal NPs in a CO/O$_2$ atmosphere were built via a Wulff construction[55]. In this scheme, the shape of a crystal at equilibrium is given by the enveloped planes, each drawn at a distance from the origin proportional to the surface tension, $\gamma_{hkl}$, of the given plane. Here, the relevant quantity is the interface tension energy, $\gamma_{hkl}^{int}$, which results from correcting the surface tension due to the interaction of a metal surface with a specific gas (or liquid).

$$\gamma_{hkl}^{int} = \gamma_{hkl} + \frac{\theta_{CO}E_{ads}(CO) + \theta_O E_{ads}(O)}{A_{at}}, \quad (1)$$

where $\theta_i$ is the surface coverage of the adsorbate $i$, CO or O. $E_{ads}$ and $A_{at}$ are the coverage-dependent energy of adsorption and the area per surface atom of the $(hkl)$ facet, respectively. In this work, we carried out DFT calculations using the BEEF-vdW functional to determine the adsorption energies of CO and O$_2$ on the various Pd and Pt facets. The calculation of adsorption–adsorption interactions, as reported previously[36–38], allows the computation of coverage dependent $E_{ads}$ values. $\theta_i$ are calculated according to the Fowler−Guggenheim adsorption isotherm model, a modification of the Langmuir isotherm to account for adsorbate–adsorbate lateral interactions, see Supplementary Note[36,37].

## Data availability

The data that support the findings of this study are available from the corresponding author upon reasonable request.

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

## Acknowledgements

This work was supported by the Singapore National Research Foundation's Competitive Research Program funding (NRF-CRP16-2015-05).

## Author contributions

S.W.C and U.M. conceived and planned the research. S.W.C. prepared the samples, performed the operando TEM experiments, and analyzed the TEM data. J.A.-R., W-Q.L., and A.G. performed the DFT calculations and theoretical modeling. All authors discussed the results. S.W.C. and U.M. prepared the manuscript with contributions and inputs from the other authors.

## Competing interests

The authors declare no competing interests.
