## [Peer Review File · Nature Communications]

Reviewers' comments:

Reviewer #1 (Remarks to the Author):

In the revised manuscript, the authors have made many changes compared to the original version. I think this manuscript might be publishable if the authors can further clarify some issues below.

(1) About the novelty, there are some ETEM studies have already reported the reshaping of Pd NPs under reaction O₂ and even N₂ conditions. Even though there is no experimental reports about the CO effect on the shape of Pd NPs, this phenomenon is expectable. The most interesting part of this work is the impact of the reshaping on the catalytic reactivity. However, the discussion of the Operando structure-reactivity relationship is not fully convincing yet.

(2) The authors argue that the increased ignition temperature with higher CO partial pressure proves the enhanced reactivity is related to the shape change. However, this statement is not so solid. The shape change at different temperature with different CO partial pressure could be different, but I did not see the corresponding theoretical modelling in the present manuscript.

(3) On the other hand, the inactivity of Pd catalysts is normally considered to be poisoning of CO that it displaces O₂ molecules. With raised temperature, the adsorption of CO could be reduced, which could explain the activation of the catalysts at elevated temperatures. At higher CO partial pressure, the temperature needed to desorb CO is increased, which could also explain the upward shift of the ignition temperature. I think the authors should give some comments about this possibility that the reactivity is merely impacted by the CO coverage but not by the structural change.

(4) To fully convince the readers that the change of the fraction of edge sites is important for the catalyst activity, I think the authors should give some theoretical calculations about the activity differences between flat surfaces and edge sites. For example, the reaction barrier calculations of CO oxidation on low index surfaces and on edged surfaces. It is important to show under the reaction conditions, the flat surfaces are indeed inactive and thus the fraction of edges sites can be really important. A micro-kinetics study would be helpful.

Reviewer #2 (Remarks to the Author):

The authors have made significant effort to address my concerns. I have no objection to publishing this work in *Nature Communications*.

Reviewers' Comments:

Reviewer #1 (Remarks to the Author):

In the revised manuscript, the authors have made many changes compared to the original version. I think this manuscript might be publishable if the authors can further clarify some issues below.

(1) About the novelty, there are some ETEM studies have already reported the reshaping of Pd NPs under reaction O₂ and even N₂ conditions. Even though there is no experimental reports about the CO effect on the shape of Pd NPs, this phenomenon is expectable. The most interesting part of this work is the impact of the reshaping on the catalytic reactivity. However, the discussion of the Operando structure-reactivity relationship is not fully convincing yet.

Our response: While we agree with the reviewer that an important aspect of the work is the correlation of structure with the catalytic activity, we emphasize that there are other scientific insights presented in this work. First, our results showing that Pd NPs revert into their inactive structure upon cooling has significant implications for the field. It implies that at least for Pd NPs and the CO oxidation reaction, **the active structure indeed cannot be determined from catalysts that are outside of reaction conditions.** To the best of our knowledge, the reversal of facet transitions as a function of temperature has not been reported in other *operando* work. **This result is also unexpected.** Although we generally assume that the working structure of a catalyst will be different from that examined under the vacuum conditions of, for example, a TEM, we still expect nanoparticles (NPs) to largely retain their high temperature structure when we decrease the temperature because kinetic processes, such as surface diffusion, become slower. Our results clearly show otherwise. This direct evidence of structure reversibility can help resolve contradictory results reported for CO binding on NPs in the existing literature. We now emphasize this point in the last line of the introduction, first paragraph of page 12 and the conclusion

We also mention here that there are key differences between our work and those mentioned by the reviewer that look at Pd NPs under the O₂ and N₂ conditions (now cited in the manuscript as 51 and 52). The fact that we included CO into the gas mix is not the only point of study in our paper. We also looked at the behavior of NPs during CO oxidation as a function of temperature/partial pressure and catalyst material in order to correlate the imaging results with reaction kinetics. **These aspects are not present in the work mentioned by the reviewer.**

Moreover, it is not clear to us from the reviewer's comments what other specific experimental evidence is needed to demonstrate the structure-reactivity correlation. Our studies involving different metal catalysts already clearly show that Pd NPs behave differently from Pt and Rh NPs in how the structure evolves with temperature, which is another observation that is novel. We also show that this difference has an impact on their low temperature activity (i.e., Pd NPs do not become active until higher temperatures) and support these results with DFT calculations. In fact, the behavior we report for Pd contrasts very well with a very recent paper on Pt from Plodinec et al. (ACS Catalysis, (2020) 10, 3183). Their detailed in situ TEM studies of Pt NPs under thermally-cycled CO oxidation conditions show that the NPs evolve continuously towards an equilibrated

structure with low index facets (consistent with our own results with Pt NPs), which is largely irreversible and lead to gradual catalyst de-activation. This is obviously different from what we saw in Pd NPs. To address this part of the reviewer's comments, we now include the movies recorded during the experiments at higher CO partial pressure to show that restructuring does indeed take place at higher temperatures to support the correlation between structure and reactivity (SI Video 5 and 6).

We also discuss the recent paper in the manuscript in page 11, paragraph 5, highlighting the key difference in the two metal systems and how it adds to our understanding of structure-reactivity correlations.

(2) The authors argue that the increased ignition temperature with higher CO partial pressure proves the enhanced reactivity is related to the shape change. However, this statement is not so solid. The shape change at different temperature with different CO partial pressure could be different, but I did not see the corresponding theoretical modelling in the present manuscript.

See response to comment 3.

(3) On the other hand, the inactivity of Pd catalysts is normally considered to be poisoning of CO that it displaces O₂ molecules. With raised temperature, the adsorption of CO could be reduced, which could explain the activation of the catalysts at elevated temperatures. At higher CO partial pressure, the temperature needed to desorb CO is increased, which could also explain the upward shift of the ignition temperature. I think the authors should give some comments about this possibility that the reactivity is merely impacted by the CO coverage but not by the structural change.

We will address both comment (2) and (3) together, since they are related. First, SI Figure 6 already shows experimental evidence that the restructuring is consistent between the two partial pressures, which is now further reinforced by SI Video 5 and 6. We have made minor changes to the wording in page 7, paragraph 1 to emphasize this point.

“Additional sets of images comparing the same NPs at the two CO to O₂ ratios of 0.5 and 1.6 (Supplementary Figure 6) indicate the same trend in structural evolution. The microcalorimetry measurements during these experiments indicate that the ignition temperature had indeed shifted upwards from...”

Second, the mechanism described in comment 3 is, in fact, our explanation for the shift in ignition temperature. In this case, we believe that the reviewer has misunderstood the order of causality that we proposed to explain our results. **In the manuscript, we do not claim an enhancement of reactivity due to the shape.** Instead, what we proposed is that the adsorption of CO molecules at low temperatures drives Pd to re-structure into an intrinsically more inactive structure (with fewer edge sites), in addition to poisoning the surface, which leads to the sharp in-active to active transition observed when CO starts to desorb from the catalysts surface (in contrast to Pt, for example, where this structural process does not happen).

Since our central premise is that the adsorption of CO molecules alters the surface structure of Pd NPs such that they become more inactive at lower temperatures, it will not be surprising that a higher partial pressure of CO enforces this process and the temperature at which we observe restructuring also shifts upwards. This mechanism is already supported by our existing calculations and will be further complemented by a discussion based on comment 3 as suggested by the reviewer at page 6, paragraph 3 of the manuscript.

(4) To fully convince the readers that the change of the fraction of edge sites is important for the catalyst activity, I think the authors should give some theoretical calculations about the activity differences between flat surfaces and edge sites. For example, the reaction barrier calculations of CO oxidation on low index surfaces and on edged surfaces. It is important to show under the reaction conditions, the flat surfaces are indeed inactive and thus the fraction of edges sites can be really important. A micro-kinetics study would be helpful.

Our response: It is not clear to us how these additional calculations requested by the reviewer add to the science presented in the paper. As mentioned in the manuscript, the correlation between undercoordinated sites and reactivity is well-accepted by the catalysis community. It has also been supported by earlier theoretical work e.g. Dahl *et al.* <https://doi.org/10.1103/PhysRevLett.83.1814>, Falsig *et al.*, <https://doi.org/10.1002/anie.200801479>. In fact, the second paper shows exactly what the reviewer wants us to model comparing the reactivity of extended metal surfaces with undercoordinated metal clusters.

We have added these two references to the manuscript and added a sentence stating that the enhanced activity of undercoordinated sites has been supported by theory (page 10, paragraph 1, line 5). We also add results from our own models showing both CO and O indeed binds stronger to the corners, consistent with calculations by Falsig *et al.* in page 11, paragraph 2.

We also deleted this sentence on page 10 of the manuscript to minimize any further confusion,

“These results also point to the corners and edges of Pd NPs as the primary active sites for CO oxidation.”

and amended the last line of the abstract to

“These insights are critical for understanding how NP surfaces can change under reaction conditions and for establishing the relation between structure and catalytic activity.”

Reviewer #2 (Remarks to the Author):

The authors have made significant effort to address my concerns. I have no objection to publishing this work in Nature Communications.

We greatly appreciate the time the reviewer had invested in reviewing this manuscript.

REVIEWERS' COMMENTS:

Reviewer #1 (Remarks to the Author):

The authors did the excellent experiments. The explanations are reasonable right now. I agree the publication of this manuscript in Nature Communications as is.